# Recent loss of the Dim2 DNA methyltransferase decreases mutation rate in repeats and changes evolutionary trajectory in a fungal pathogen

Mareike Möller[1,2¤]*, Michael Habig[1,2], Cécile Lorrain[1,2], Alice Feurtey[1,2], Janine Haueisen[1,2], Wagner C. Fagundes[1,2], Alireza Alizadeh[3], Michael Freitag[4], Eva H. Stukenbrock[1,2]*

**1** Environmental Genomics, Christian-Albrechts University, Kiel, Germany, **2** Max Planck Institute for Evolutionary Biology, Plön, Germany, **3** Department of Plant Protection, Faculty of Agriculture, Azarbaijan Shahid Madani University, Tabriz, Iran, **4** Department of Biochemistry and Biophysics, Oregon State University, Corvallis, OR, United States of America

¤ Current address: Department of Biochemistry and Biophysics, Oregon State University, Corvallis, Oregon, United States of America

* estukenbrock@bot.uni-kiel.de (EHS); moellmar@oregonstate.edu (MM)

**Data Availability Statement:** Sequencing raw reads (FASTQ files) of all bisulfite and genomic data and the SMRT genome assemblies generated

## Abstract

DNA methylation is found throughout all domains of life, yet the extent and function of DNA methylation differ among eukaryotes. Strains of the plant pathogenic fungus *Zymoseptoria tritici* appeared to lack cytosine DNA methylation (5mC) because gene amplification followed by Repeat-Induced Point mutation (RIP) resulted in the inactivation of the *dim2* DNA methyltransferase gene. 5mC is, however, present in closely related sister species. We demonstrate that inactivation of *dim2* occurred recently as some *Z. tritici* isolates carry a functional *dim2* gene. Moreover, we show that *dim2* inactivation occurred by a different path than previously hypothesized. We mapped the genome-wide distribution of 5mC in strains with or without functional *dim2* alleles. Presence of functional *dim2* correlates with high levels of 5mC in transposable elements (TEs), suggesting a role in genome defense. We identified low levels of 5mC in strains carrying non-functional *dim2* alleles, suggesting that 5mC is maintained over time, presumably by an active Dnmt5 DNA methyltransferase. Integration of a functional *dim2* allele in strains with mutated *dim2* restored normal 5mC levels, demonstrating *de novo* cytosine methylation activity of Dim2. To assess the importance of 5mC for genome evolution, we performed an evolution experiment, comparing genomes of strains with high levels of 5mC to genomes of strains lacking functional *dim2*. We found that presence of a functional *dim2* allele alters nucleotide composition by promoting C to T transitions (C→T) specifically at CpA (CA) sites during mitosis, likely contributing to TE inactivation. Our results show that 5mC density at TEs is a polymorphic trait in *Z. tritici* populations that can impact genome evolution.

in this study are available online at Sequence Read Archive (SRA) under BioProject ID PRJNA614493. Previously published genome assemblies are available under BioProject PRJEB33986, PRJNA638605, PRJNA639021 (Zpa63), PRJNA638553 (Zb87), PRJNA638515 (Zp13), PRJNA638382 (Za17), https://doi.org/10.5281/zenodo.3820378, PRJNA414407 (Zt05 and Zt10). The genome sequence of the reference isolate IPO323 (Zt09) is available at: http://genome.jgi.doe.gov/Mycgr3/Mycgr3.home.html. In planta RNA-seq datasets for Z. tritici Zt09 and Zt10 are available at NCBI Gene Expression Omnibus with the accession number GSE106136. In vitro RNA-seq data for Zt09 is available in the NCBI Sequence Read Archives SRX5578740 and SRX5578741.

**Funding:** Research in the lab of EHS is supported by the State of Schleswig-Holstein, the Max-Planck-Gesellschaft and CIFAR. Research in the lab of MF is supported by National Science Foundation (NSF) grant (MCB1818006). MM is supported by the Deutsche Forschungsgemeinschaft (DFG, MO 3755/1-1). The funders had no role in study design, data collection and analysis, decision to publish, or preparation of the manuscript.

**Competing interests:** The authors have declared that no competing interests exist.

## Author summary

Cytosine DNA methylation (5mC) is known to silence transposable elements in fungi and thereby appears to contribute to genome stability. The genomes of plant pathogenic fungi are highly diverse, differing substantially in transposon content and distribution. Here, we show extensive differences of 5mC levels within a single species of an important wheat pathogen. These differences were caused by inactivation of the DNA methyltransferase Dim2 in the majority of studied isolates. Presence of widespread 5mC increased point mutation rates in regions with active or mutated transposable elements during mitosis. The mutation pattern is dependent on the presence of Dim2 and resembles a mitotic version of Repeat-Induced Point mutation (RIP). Thus, loss of 5mC may represent an evolutionary trade-off offering adaptive potential at the cost of transposon control.

## Introduction

DNA methylation is an important process for epigenetic regulation, and functions range from dynamic control of gene expression to transposon silencing and the maintenance of genome integrity [1,2]. Although DNA methylation has been detected on both cytosines and adenines in eukaryotes, cytosine DNA methylation (5mC) has been the focus of most studies so far. In mammals, cytosine methylation is mainly found in "CpGs" (cytosine followed by guanine; here abbreviated CG), while non-CpG methylation (CHG or CHH, where H is any nucleotide other than G) is commonly detected in plants and fungi [3,4]. Various DNA methyltransferases (DNMTs) are involved in the establishment and maintenance of DNA methylation but the distribution and number of enzymes involved is highly variable in different kingdoms [5]. In mammals and plants, enzymes of the DNMT1/MET1 class are maintenance methyltransferases that detect hemi-methylated DNA sequences, for example after replication [6–8]. *De novo* methyltransferases, like DNMT3a and DNMT3b in mammals [9] and DRM in plants [10] act on sequences that are free of methylation, presumably by recognition of specific motifs or patterns [9,11]. Although DNMTs are often classified as maintenance or *de novo* enzymes their function is not necessarily limited to one or the other [12,13].

In fungi, four classes of DNMTs have been identified. Basidiomycetes have DNMT1 homologs that appear to be classical maintenance DNMTs [4,14]. In the ascomycete *Neurospora crassa*, DNA methylation is mediated by a single enzyme, DIM-2 [15]. While the conserved DIM2 class of enzymes shows limited sequence similarity to DNMT1/MET1 maintenance DNMTs, it is a class specific to fungi [16]. Other fungal proteins resembling DNMT1 include *Ascobolus immersus* Masc1 and *N. crassa* RID [17,18], involved in "Methylation Induced Premeiotically" (MIP) or "Repeat-Induced Point mutation" (RIP), respectively [17,19]. DNMT5 enzymes constitute a more recently discovered class of maintenance DNMTs in fungi [20,21]. So far, the presence of 5mC in fungi has been mainly associated with repetitive DNA, suggesting a role in genome defense by silencing of transposable elements (TEs). There is little or no evidence that 5mC is also found in coding sequences to influence gene expression [22], although there are examples of promoter methylation [23,24].

A previous study demonstrated amplification of *dim2*, a gene encoding the homolog of *N. crassa* DIM-2, which was predicted to result in mutation and thus inactivation by RIP in the genome of the plant pathogenic fungus *Zymoseptoria tritici* [25]. For example, the genome of the reference isolate IPO323 carries 23 complete or partial, nonfunctional copies of *dim2*; all alleles show signatures of RIP, namely numerous C:G to T:A transition mutations. Consequently, 5mC was not detected by mass spectrometry in IPO323 [25]. However, the

amplification of *dim2* must have occurred recently as two closely related sister species of *Z. tritici*, *Zymoseptoria ardabiliae* and *Zymoseptoria pseudotritici* were shown to carry a single intact *dim2* gene and have 5mC [25].

Population genomic analyses have revealed high levels of genetic variation within and between populations of *Z. tritici* [26–28]. This variation is generated by high mutation and recombination rates and by extensive gene flow. Notably, a dynamic landscape of transposable elements in this fungus has been associated with rapid evolution, not only in *Z. tritici*, but also in sister species [29,30]. Moreover, a recent study demonstrated a pervasive effect of recurrent introgression between closely related species of *Zymoseptoria* [31]. These interspecific hybridization events are evident from the presence of highly diverged alleles that are maintained over evolutionary time within species.

By genome analyses of multiple *Z. tritici* isolates from the center of origin of the pathogen, the Middle East, we discovered several *Z. tritici* isolates with an intact *dim2* gene. Our findings suggest that the loss of *de novo* 5mC not only occurred very recently but is an ongoing process and polymorphic trait in *Z. tritici*. Here, we address the evolution and function of *dim2* in the *Zymoseptoria* species complex. We show that the presence of an intact and functional *dim2* in some *Z. tritici* isolates corresponds to widespread 5mC in TEs. Integration of a functional *dim2* allele into a *dim2*-deficient background restores methylation of previously non-methylated regions. Using comparative genomics and experimental evolution we show that loss of *dim2* and thus 5mC affects nucleotide composition of TEs by influencing mutation rates at CA sites.

## Results

### The *dim2* DNA methyltransferase gene is functional in several *Z. tritici* isolates

Based on previous findings reporting a recent inactivation of *dim2* in the *Z. tritici* reference isolate IPO323 [25] we carefully inspected the genomic locus in different isolates to characterize the extent of mutational events in different genomes. We identified a non-truncated *dim2* gene in an Iranian *Z. tritici* isolate, Zt10, and set out to investigate its recent evolution in the *Zymoseptoria* species complex in a much larger collection of genome sequences than previously [25] available. We used BLAST to search for the sequence of *dim2* in 22 high-quality assemblies obtained by SMRT sequencing of *Z. tritici* (Fig 1) [32–35], as well as 17 genomes of *Z. ardabiliae*, and nine genomes of *Z. brevis* (S1 Table), both closely related sister species of *Z. tritici* and considered to be endemic to Iran. We detected a single *dim2* homolog in each of the *Z. ardabiliae* and *Z. brevis* genomes but found multiple mutated and presumably non-functional copies of *dim2* in 17 out of 22 *Z. tritici* genomes (Fig 1). Five *Z. tritici* isolates contained one presumably functional, non-mutated copy of *dim2*, often in addition to multiple mutated, non-functional copies, except for isolate Zt469, which contained one functional *dim2* gene and no additional copies. For the purpose of this study, we consider *dim2* alleles intact and 'functional' when they do not contain pre-mature stop codons or frameshift mutations, even though some SNVs may generate missense mutations. In contrast, 'non-functional' alleles contain numerous mutations including pre-mature stop codons that are predicted to abrogate protein function. Actual methylation activity was tested in selected strains by assaying cytosine methylation *in vivo*.

Zt469 was isolated from *Aegilops* sp., all other isolates were collected from wheat. Four of five isolates with functional *dim2* are from Iran (Zt10, Zt11, Zt289, Zt469) [32,36], and one is from Tunisia (TN09, collected from durum wheat) [32]. We further tested whether presence of a non-mutated *dim2* allele is commonly found among Iranian isolates. Twelve additional

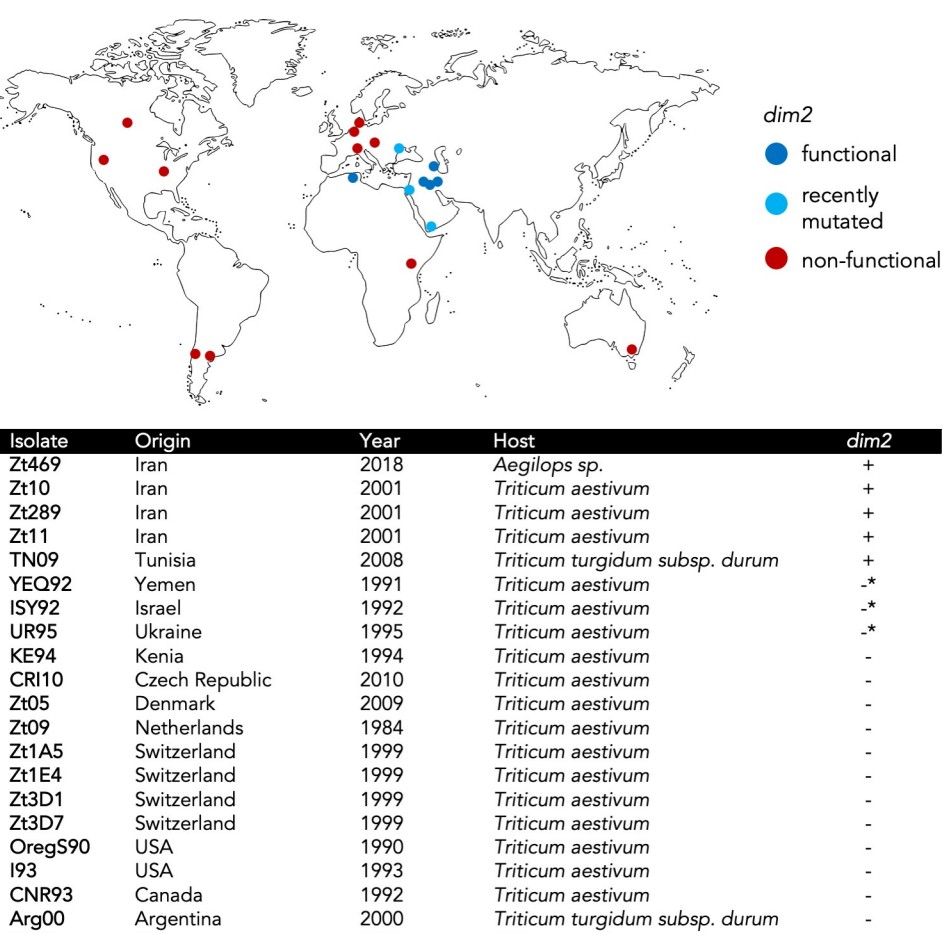

Fig 1. Overview of sampling location, year, host plant, and presence of a functional *dim2* DNA methyltransferase gene of *Z. tritici* isolates analyzed in this study. *Aegilops* sp., wild grass in Triticeae; *Triticum aestivum*, bread wheat; *T. turgidum durum*, Durum wheat. *non-functional but only recently mutated native *dim2* gene.

isolates from Iran (S1 Table) were assayed by PCR and Sanger sequencing and in all cases, we confirmed presence of an intact *dim2* allele. A region of chromosome 6 of IPO323 had been previously identified as the native *dim2* locus; all additional, non-functional copies are located in subtelomeric regions embedded in repetitive DNA [25]. Consistent with this idea, we found that all non-mutated *dim2* genes were located at the predicted native locus on chromosome 6 (Fig 2A and S2 Table). Throughout this study, we refer to *dim2* in the original locus on chromosome 6, functional or non-functional, as the 'native' gene or allele; *dim2* copies outside the native locus are referred to as 'non-native' or 'additional' alleles.

## Analysis of *Z. tritici dim2* alleles reveals multiple cycles of recombination and inactivation

Previously, gene amplification of the native *dim2* allele followed by RIP had been suggested as the most likely cause for the apparent lack of 5mC in the *Z. tritici* IPO323 genome [25,37]. We here conducted a detailed analysis of mutations within native and non-native *dim2* genes in the *Z. tritici* genomes and found some unexpected patterns.

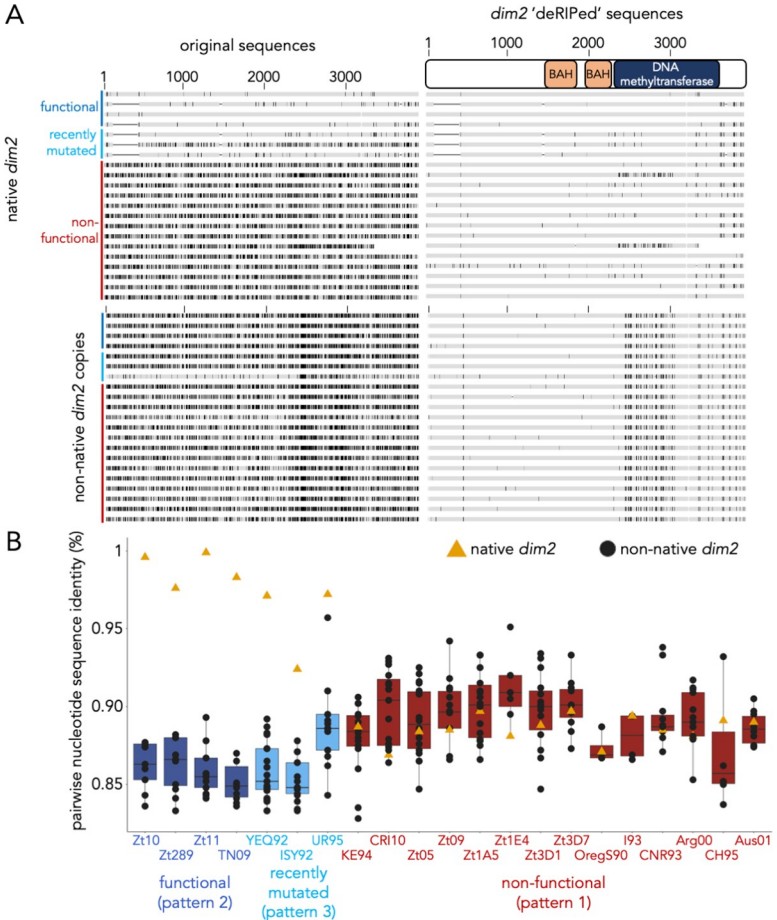

**Fig 2. Sequence comparisons of *dim2* alleles in *Z. tritici* isolates. (A)** Mutations of *dim2* in original and 'deRIPed' alleles identified by alignment to the functional allele of Zt469. Shown are all native alleles (located on chromosome 6) and one representative non-native, additional copy per isolate. Native alleles (functional and non-functional) lack mutations in the DNA methyltransferase domain that are present in all non-native alleles, except for isolates OregS90 and CRI10. **(B)** Pairwise sequence comparison of the native, intact *dim2* allele from Zt469 with all full-length native and additional *dim2* sequences (see S2 Table). No additional *dim2* copies were detected in the genome of Zt469 (from *Aegilops* sp.) whereas all other isolates (from *Triticum* spp.) have additional copies of *dim2*. Non-native, additional copies (black dots) in the isolates with functional or recently mutated *dim2* (except for UR95) showed higher sequence diversity compared to those with inactivated *dim2*, where alleles at the native locus (yellow triangle) are amongst the more diverged copies. This supports the idea that this is the ancestral, now heavily mutated allele of *dim2*. Alleles or isolates are arranged in the same order in panels A (top to bottom) and B (left to right).

As previously reported, non-functional copies of *dim2* contain numerous transition mutations as a consequence of RIP, thus to compare sequences of native and additional (non-native) copies we "deRIPed" mutant copies by reverting C→T and G→A transitions using the functional *dim2* from Zt469 as reference, because it likely represents one ancestral allele. We found that all non-native alleles share a specific pattern of transition and, surprisingly, transversion mutations in the DNA methyltransferase domain that are not found in the vast majority of native *dim2* genes (Figs 2A and S1A and S2). Transversions are not a consequence of RIP [38,39], thus a mixture of transitions, transversions, and deletions suggests that *dim2* was mutated not simply as a result of amplification of the native gene followed by widespread RIP, as originally proposed [25]. Rather, integration of a different *dim2* allele (hereafter called 'transversion allele') of unknown origin into subtelomeric regions is likely to have resulted in

subsequent gene amplification, RIP and presence of multiple non-functional, non-native *dim2* copies. The presence of the same transversion mutations in all additional copies indicates that this allele may have resulted from a single integration event in an ancestral *Z. tritici* isolate. In support of this hypothesis, we found that the flanking regions of the additional, but not the native, copies share homology and are annotated as retrotransposons, suggesting that integration and amplification of non-native copies may have been TE-mediated. The TEs that are found in the flanking regions of most of the additional *dim2* copies belong to the LINE family.

While this idea explains events at subtelomeric integrations sites, we also identified two isolates in which the native *dim2* gene on chromosome 6 shares the specific transversion mutations in the DNA methyltransferase domain, OregS90 and CRI10 (from the USA and Czech Republic, respectively) (Fig 2A). The native OregS90 *dim2* is truncated and surrounded by TEs and the native *dim2* of CRI10 lacks mutations in the 3' region that are present in all additional, but not native, copies.

Five native *dim2* genes have a shared deletion in the 5' region that is not found in any of the additional copies, and a 27-bp insertion at the end of the reading frame. The 5' deletion results in a 99–109 amino acid in-frame deletion and not in any known functionally important domains, while the ill-conserved C-terminus is extended by nine amino acids (S3 Fig). These results suggest that distinct functional *dim2* alleles are present, and given the previously identified signatures of introgression in *Z. tritici* genomes [31] we consider that these different *dim2* alleles at the native locus may reflect separate integration events that have been directed by sequence identity. In summary, our analyses revealed that gene amplification followed by RIP alone is insufficient to explain the distribution of variant *dim2* alleles in *Z. tritici* isolates.

To further investigate the evolutionary history of *dim2* in *Z. tritici* isolates, we compared the sequence identity of the non-mutated *dim2* gene from Zt469 to all other, native and additional, copies of *dim2* in the 22 *Z. tritici* genomes (Fig 2B). Gene amplification followed by RIP predicts that by accumulating transition mutations all copies should become quite similar to each other; RIP becomes less efficient or stops when sequence identity between alleles drops to ~80% [38]. Instead, our analyses revealed that 18/21 isolates showed two distinct patterns, the first matching the prediction because isolates lacking functional *dim2* showed higher levels of DNA sequence identities (85–95%) between native and additional alleles (Fig 2B and S2 Table; red, "non-functional", pattern 1). Here, all copies have suffered repeated RIP cycles, and all copies are non-functional. When, however, a functional, native *dim2* allele is present, the additional copies show lower DNA sequence identities (84–88%), likely reducing RIP efficiency and preventing the functional allele from being inactivated. The functional alleles are almost identical (>96%) to the Zt469 allele (Fig 2B and S2 Table; dark blue, "intact", pattern 2; Zt10, Zt289, Zt11, TN09). This second pattern was unexpected; the most parsimonious explanation is replacement of non-functional alleles at the native locus with novel *dim2* alleles, perhaps by recombination with wild grass infecting isolates, such as Zt469. Introgression events between *Zymoseptoria* species have recently been described [31] and wild grass infecting *Z. tritici* isolates, sister species and wheat infecting *Z. tritici* populations co-exist in the same geographical regions in the Middle East [40].

We discerned a third group of three isolates, which suggests that repeated cycles of transposon- or recombination-mediated amplification of *dim2* alleles are still ongoing (Fig 2B and S2 Table; light blue, "recently inactivated", pattern 3; YEQ92, ISY92, UR95). YEQ92 and ISY92 contain non-functional *dim2* alleles at the native locus that are still similar (96 and 92%, respectively) to the functional *dim2*, suggesting a more recent inactivation of *dim2* in these isolates. All additional, subtelomeric copies have low sequence identities as in the second pattern described above. UR95 has two recently mutated *dim2* alleles, one at the native locus (Zt469 allele) and one in the subtelomeric region on chromosome 8 (transversion allele); both are

non-functional alleles but contain relatively few mutations and are still similar to the functional *dim2* allele (>90%) suggesting recent integration and inactivation. In summary, our findings suggest that there are multiple functional alleles of *dim2* at the native locus present in the *Z. tritici* population. One group of alleles is derived from the Zt469 allele, a second group contains the "fingerprint" of the transversion allele when compared to the Zt469 allele and a third group contains the 5' deletion. All alleles exhibit some degree of variation in terms of synonymous, non-synonymous mutations, and INDELs but none of these mutations affect sites that are known to be essential for protein function indicating that all of these alleles may likely encode functional proteins (S3 Fig).

## Integration of a functional *dim2* allele at the native locus restores 5mC in Zt09

To assess the role of *dim2* in DNA methylation we generated mutants in which we either restored or deleted *dim2*. We integrated the functional *dim2* gene of the Iranian Zt10 isolate into the native *dim2* locus of Zt09 (derived from reference isolate IPO323) by selecting for hygromycin resistance, conferred by the *hph* gene, generating the strain Zt09::*dim2*. In Zt10, we replaced the functional *dim2* gene with *hph*, generating the strain Zt10Δ*dim2*. Integration or deletion of the *dim2* alleles were verified by Southern analyses (S4 Fig). To test whether *dim2* affects growth or infection, we compared the mutant and respective wild-type strains under different *in vitro* conditions as well as during host infection. We did not detect any differences between wild-type and mutant strains *in vitro* (S5 Fig) but integration of the functional *dim2* in Zt09 resulted in a small but significant reduction in necrosis and asexual fruiting body formation (measured as number of asexual fruiting bodies, pycnidia, per leaf area) *in planta* (S6 Fig). We therefore conclude that presence of *dim2* is dispensable under the tested *in vitro* conditions and that re-integration of *dim2* into the Zt09 background decreases virulence *in planta*. We note that restoring DNA methylation in Zt09 after absence of *dim2* for likely thousands of generations does not necessarily reflect the influence that DNA methylation may have in *Z. tritici* under normal conditions. Deletion of *dim2* in Zt10 did not have significant immediate impact on virulence; this needs to be further addressed in long term studies.

To validate the presence of 5mC and to compare patterns between isolates and mutants, we performed whole-genome bisulfite sequencing (WGBS). We sequenced three replicates of each wild-type isolate (Zt09 and Zt10) and mutant strain (Zt09::*dim2* and Zt10Δ*dim2*). As a control for the bisulfite conversion rate, we added Lambda DNA to each sample; based on this, the bisulfite conversion rate was >99.5%. To confirm the WGBS sequencing results, we performed an additional independent bisulfite conversion, amplified target regions by PCR, cloned and sequenced the fragments, and we performed Southern blot analyses using 5mC-sensitive and -insensitive restriction enzymes (S7 Fig).

We detected high numbers of methylated cytosines in isolates with functional *dim2* alleles, i.e., the wild-type Iranian isolate Zt10 and Zt09::*dim2*. Surprisingly, we also detected low levels of 5mC in the absence of *dim2*, i.e., in Zt10Δ*dim2* and Zt09, which had been missed by mass spectrometry in earlier studies [25]. While on average ~300,000 sites in Zt10 and ~400,000 sites in Zt09::*dim2* showed 5mC in the presence of *dim2*, only ~12,000 (Zt09) and ~40,000 (Zt10Δ*dim2*) 5mC sites were found when *dim2* was absent (Fig 3A). In all strains, 5mC was restricted to previously annotated TEs and adjacent non-coding regions (Fig 3A). We did not find evidence for gene body or promoter methylation, except for few TE-derived genes. By correlating the 5mC data with previously published histone methylation maps [41] we found that 5mC co-localizes with H3K9me3 marks (Fig 3B). This finding is consistent with observations in *N. crassa*, *C. neoformans*, and *Arabidopsis thaliana* [21,42–45].

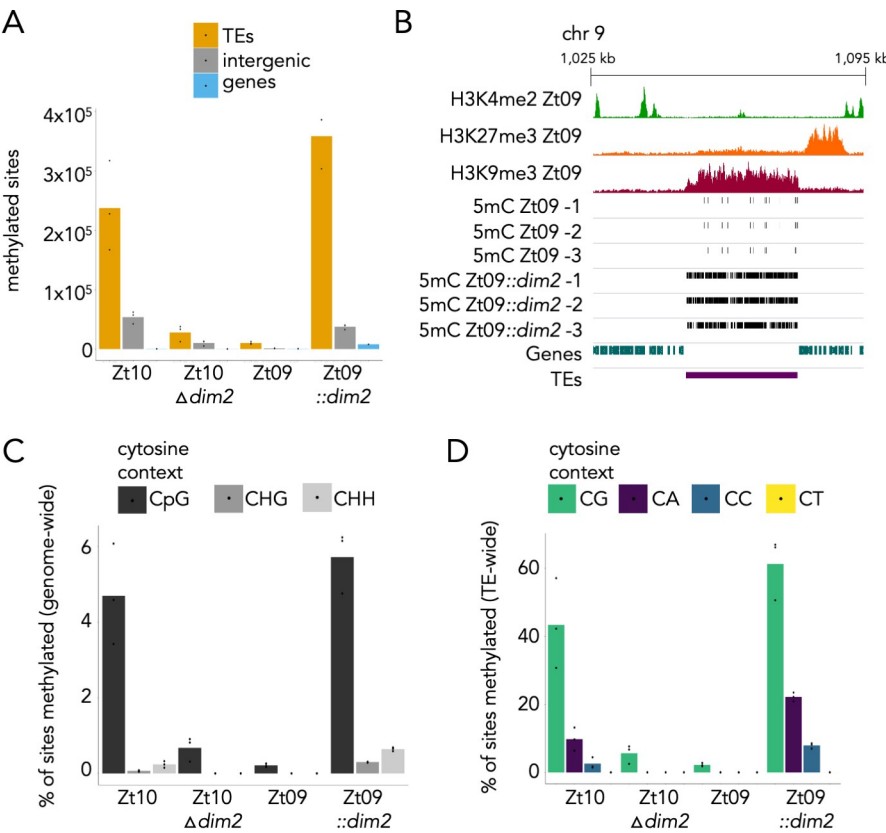

**Fig 3. Localization and site preferences of cytosine methylation in *Z. tritici* genomes in presence and absence of *dim2*. (A)** Number of 5mC methylated sites detected by whole genome bisulfite sequencing (WGBS) in TEs, intergenic regions, or genes. The vast majority of methylated sites are localized in TEs followed by intergenic regions and TE-derived genes. **(B)** 5mC co-localizes with H3K9me3 and TEs in Zt09. Shown are ChIP-seq tracks for H3K4me2, H3K27me3 and H3K9me3 [41], and 5mC sites detected in a representative region on chromosome 9 in the genomes of Zt09 and Zt09::*dim2* (three replicates). **(C)** Genome-wide 5mC levels in the different isolates and respective mutants. 5mC levels are higher in the presence of a functional *dim2*, and *dim2* is required for all non-CG methylation. **(D)** 5mC levels and site preferences in TEs. CG sites are preferred, followed by CA, and CC. 5mC is almost completely absent from CTs.

We next analyzed the specific context of DNA methylation. In strains without functional *dim2*, we found 5mC exclusively in CG contexts (genome wide ~0.2% in Zt09 and ~1% in Zt10Δ*dim2*; percentage based on all Cs in the genome*)*. In the presence of *dim2*, cytosines in CG as well as CHG and CHH contexts were methylated. Genome wide, ~5 or 6% of CGs and <1% of CHGs or CHHs were methylated in Zt10 or Zt09::*dim2*, respectively (Fig 3C). In sequences annotated as TEs, however, up to 60% of all CGs were methylated (Fig 3D). Methylation of CGs occurred most frequently (~43%, ~61%), but we also detected methylation at CAs (~10%, ~22%), and to lesser extent at CCs (~3%, ~8%), in Zt10 or Zt09::*dim2*, respectively. At CTs, there was hardly any detectable methylation (<0.01%; Figs 3D and S8). This was surprising, as CTs are the most abundant CH dinucleotides in TEs while CAs are the least abundant sites in TEs of both Zt09 and Zt10 (S3 Table). This suggests a strong site-specific preference of Dim2 for CAs but not CTs in *Z. tritici*. We furthermore calculated methylation levels (number of 5mC reads/position) and found that 5mC levels and overlap of 5mC sites between replicates are highest in Zt09, where only very few CpG sites are maintained (S9 Fig). Zt10Δ*dim2* shows high variability of 5mC site positions and methylation levels between the three replicates (independent transformants) indicating that 5mC sites are nearly randomly

and progressively lost after *dim2* deletion. Zt10 and Zt09::*dim2* both have comparable methylation levels that are consistent between replicates (S9 Fig).

Upon integration of a functional *dim2* allele in Zt09 we found that overall 5mC levels were higher when compared to Zt10 (compare Fig 3C, left- and right-most panels) and CG methylation levels in Zt10Δ*dim2* were higher compared to Zt09 (compare Fig 3C, two middle panels). One possible explanation for higher methylation levels in the Zt09::*dim2* strain is that upon integration of *dim2*, a high number of sites are methylated but then reduced by selective conditions. This may occur after several rounds of infection and prolonged growth until a balanced level of 5mC sites is achieved, as found in Zt10. It is also possible that methylation levels differ between isolates and may depend on the abundance of different TE families that vary in terms of methylation density. Our findings suggest that in the absence of the *de novo* DNA methyltransferase Dim2, 5mC levels decrease significantly over relatively short evolutionary time spans ($< 12,000$ years, after speciation of *Z. tritici* [36]).

## DNA methylation can be maintained in absence of *dim2*

We next asked how 5mC can persist in the genome of *Z. tritici* in the absence of a functional *dim2* gene. The presence of CG methylation in isolates without *dim2* suggested the presence of an additional DNMT. Thus, we searched for putative DNMT coding regions in the genomes of *Z. tritici*, *Z. ardabiliae*, and *Z. brevis* with the conserved DNMT domain of Dim2 as query in a BLASTp search. In all genomes, we found two additional predicted DNMTs (S11 Fig). One is similar to Dnmt5 of *Cryptococcus neoformans* [20] (*dnmt5*, Zt_chr6_00685), the other is a homolog of *N. crassa* RID [17] (*rid*, Zt_chr5_00047; manually corrected gene annotation) [46]. Genes for Dnmt5 and Rid are present in all *Zymoseptoria* spp. genomes we analyzed. The *dnmt5* alleles are highly conserved and show very little inter- or intraspecies diversity (S1B Fig). In contrast, *rid* shows an exceptionally high inter- and intraspecies diversity with three highly distinct alleles present in *Zymoseptoria* spp. (S1C Fig). This diversity may stem from introgression events between *Zymoseptoria* species as recently described for highly variable regions in the genome of *Z. tritici* [31]. However, presence of the different *rid* alleles does not correlate with presence of a functional *dim2* gene in *Z. tritici*.

RID and its homolog Masc1 in *Ascobolus immersus* are involved in genome defense during pre-meiosis [17,18]. Although Masc1 is presumed to be a *de novo* DNMT acting during the sexual cycle of *A. immersus* [18], there are still no enzyme activity data available for either Masc1 or RID. In contrast, Dnmt5 has been characterized as a 5mCG maintenance DNMT in *C. neoformans* [20,21] and is thus the most likely candidate for maintenance of 5mC at CG sites that we observed here. *Dnmt5* is moderately expressed during infection of wheat in both, Zt09 and Zt10, while the expression levels of *rid* are very low consistent with its known role in pre-meiotic, and not mitotic, stages (S10 Fig) [17]. Further studies will address the role of Dnmt5 in maintenance and impact on 5mC in *Z. tritici*.

## Presence of *dim2* impacts nucleotide composition and activity of TEs

DNA cytosine methylation is correlated with increased C→T mutation rates [47,48]. In *Z. tritici* and most other fungi [22], 5mC is enriched in TE sequences. To evaluate whether the presence of *dim2* impacts sequence composition of TEs, we analyzed TE sequences in isolates with and without a functional *dim2* gene. Therefore, we extracted TE sequences, calculated dinucleotide frequencies relative to the TE content for each genome (S3 Table), and visualized the difference in relative frequencies of specific dinucleotides in TEs in the different isolates by correspondence analysis. We found that isolates lacking functional *dim2* have more CA/TG sites within TEs compared to isolates with functional *dim2* (Fig 4A and 4B). Conversely,

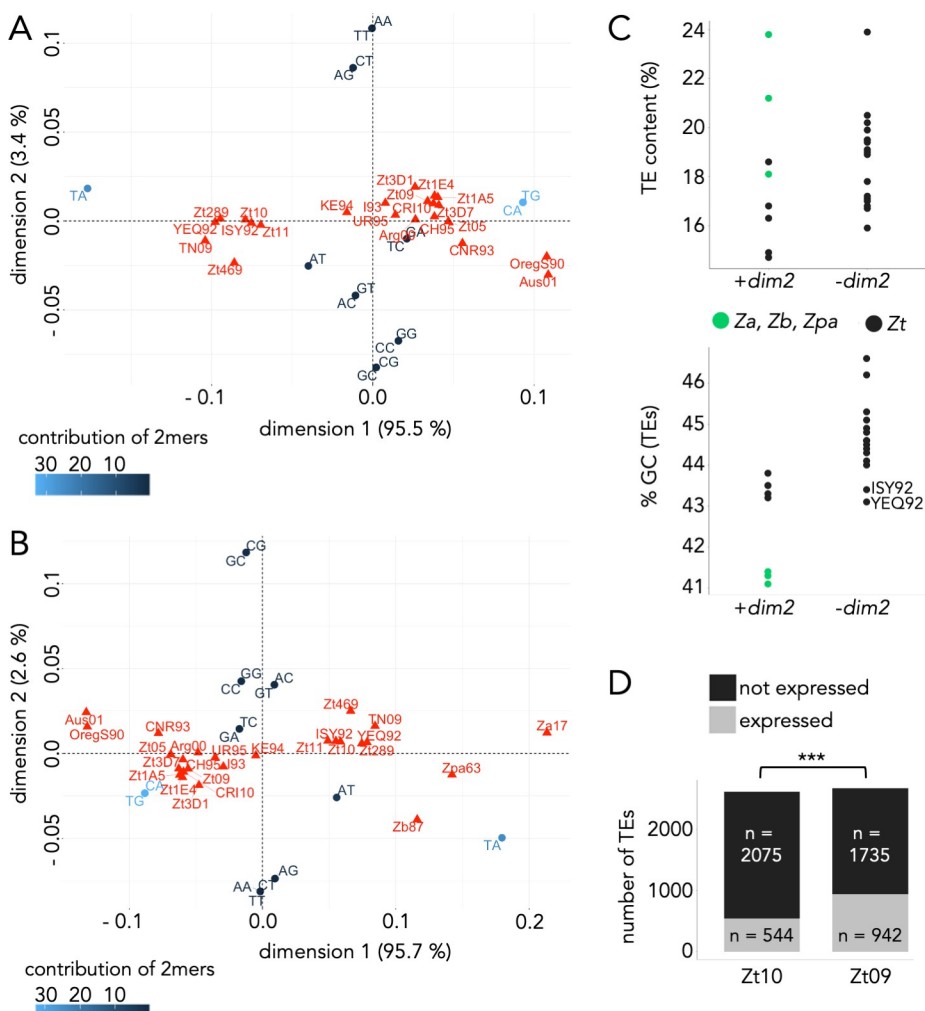

**Fig 4. Impact of Dim2 on nucleotide composition and TE activity. (A)** and **(B)** Correspondence analyses showing differences in dinucleotide (2mer) frequencies in TEs of different *Z. tritici* isolates **(A)** and other members of the *Zymoseptoria* species complex **(B)**. TA and CA/TG dinucleotide frequencies have the highest impact on the observed differences between isolates and species, indicated by their relative positions on the dimension 1 *versus* dimension 2 plot. Isolates (even from different species) containing a functional *dim2* (Zt10, Zt289, Zt11, Zt469, TN09, Zb87, Za17, Zpa63) and recently mutated *dim2* (ISY92 and YEQ92) have higher TA content and cluster together, while isolates without functional *dim2* have higher CA/TG frequencies in TE sequences. **(C)** Comparison of TE and GC content between different *Z. tritici* isolates and *Zymoseptoria* species with and without functional *dim2*. **D)** Number of expressed TEs during the time course of wheat infection is significantly higher in *Z. tritici* isolate Zt09 (non-functional *dim2*) compared to Zt10 (functional *dim2*) (Fisher's Exact Test for Count Data, *p*-value < 2.2 x 10$^{-16}$).

isolates with functional *dim2* contain more TA sites in TEs (Fig 4A and 4B). Two isolates with recently inactivated *dim2*, ISY92 and YEQ92, show the same pattern as isolates with functional *dim2* but UR95, with two recently inactivated copies, groups with isolates lacking functional *dim2*. Based on the presence of additional *dim2* copies with relatively high sequence identities in UR95 (Fig 2B), we propose that integration of a functional *dim2* may lead to rapid inactivation by RIP. Thus, grouping of UR95 with the 'non-functional' isolates can be explained by limited Dim2 activity before inactivation of the gene by RIP, a prediction that will be tested in future studies. CA/TG sites are the least frequent sites in TEs, while TA sites are the most abundant. We observed the same pattern (i.e., increased number of TA sites) when we

included the sister species *Z. ardabiliae* (Za17) and *Z. brevis* (Zb87) and the more distantly related barley pathogen, *Z. passerinii* (Zpa63), all of which contain a single, functional *dim2* (Fig 4B and S3 Table). This pattern is only detectable in TEs, not in the rest of the genome (S3 Table).

Accelerated mutation rates in TEs may impact TE activity, so we next determined the TE content of the various *Z. tritici* isolates and compared it to that observed in other *Zymoseptoria* species (Fig 4C). Isolates with functional *dim2* had slightly lower TE content than those lacking *dim2* (mean of ~16.2% *vs.* ~18.7%, respectively; Fig 4C) but TE content in other *Zymoseptoria* species, all with functional *dim2*, was higher than in many *Z. tritici* strains. Thus, there is no simple or clear correlation between TE content and *de novo* 5mC activity. GC content of TEs, however, is considerably lower in isolates with functional *dim2* and two isolates with recently mutated *dim2*, ISY92 and YEQ92, further indicating that Dim2 impacts nucleotide composition (Fig 4C).

The TE content of a genome does not necessarily reflect the activity of TEs, as these sequences may be relics, i.e., mutated and thus inactivated TEs. We therefore assessed whether TEs in Zt09 and Zt10 contained annotated, transposon-related genes as a putative indicator for the presence of active TEs. In Zt09, 56 genes completely overlap (>90% of the sequence) with annotated TEs. Among those, we identified 30 transposon- or virus-related genes encoding for endonucleases, transposases, ribonucleases, and reverse transcriptases, or genes encoding virus-related domains (S4 Table). In contrast, we did not detect any fungal genes overlapping TEs in Zt10. We did find homologs of some of the transposon-associated genes of Zt09 in the genome of Zt10, but they all contained numerous transitions resulting in mis- and nonsense mutations. This indicates that Zt09 contains active TEs whereas homologous elements are inactivated in Zt10.

Lastly, we compared the expression of TEs in Zt09 and Zt10 during the time course of infection by analyzing previously published RNA-seq data [35]. We computed transcripts per million (TPM) originating from TEs in both the biotrophic and necrotrophic phases of infection and considered a TE expressed when TPM was > 0. While ~21% of TEs produced RNA in Zt10, ~35% of TEs were transcribed in Zt09 (Fig 4D) indicating that silencing of TE loci is less pronounced in Zt09 where *dim2* is non-functional.

## Dim2 promotes C→T transitions at CAs during mitosis

The difference between CA/TG and TA site abundance in genomes with and without functional *dim2* suggests that the mutation rate, specifically for C→T transitions, is affected by the presence of 5mC. To address whether Dim2 plays a role in promoting these mutations, we conducted a one-year evolution experiment that included the Iranian isolate Zt10 (functional *dim2*) and the reference isolate IPO323 (non-functional *dim2*, isolate Zt09 is a derivate of IPO323). This was a mutation accumulation experiment on cells that were dividing by mitosis without competition or selection. Each ancestral isolate was propagated as 40 independent replicates. After 52 weeks (corresponding to ~1,000 mitotic cell divisions) the genomes of 40 evolved replicate strains per isolate and the progenitor isolates were sequenced to map SNPs. We found that the mutation rate in Zt10 was more than tenfold higher than in IPO323, on average ~170 *vs.* ~13 mutations per replicate (Fig 5A and S5 Table). The vast majority of mutations in Zt10 (> 95%) occurred in 5mC regions (S6 Table) and > 95% of all mutations were C→T transitions, while only ~32% of all mutations in IPO323 were C→T transitions. In Zt10, > 98% of the C→T transitions occurred at CAs (*vs.* ~21% in IPO323) (Fig 5B). This was a surprise, because the majority of methylated sites are at CGs. Our findings suggest the presence of a *dim2*-dependent mechanism that specifically targets and mutates CA/TG but not CGs in

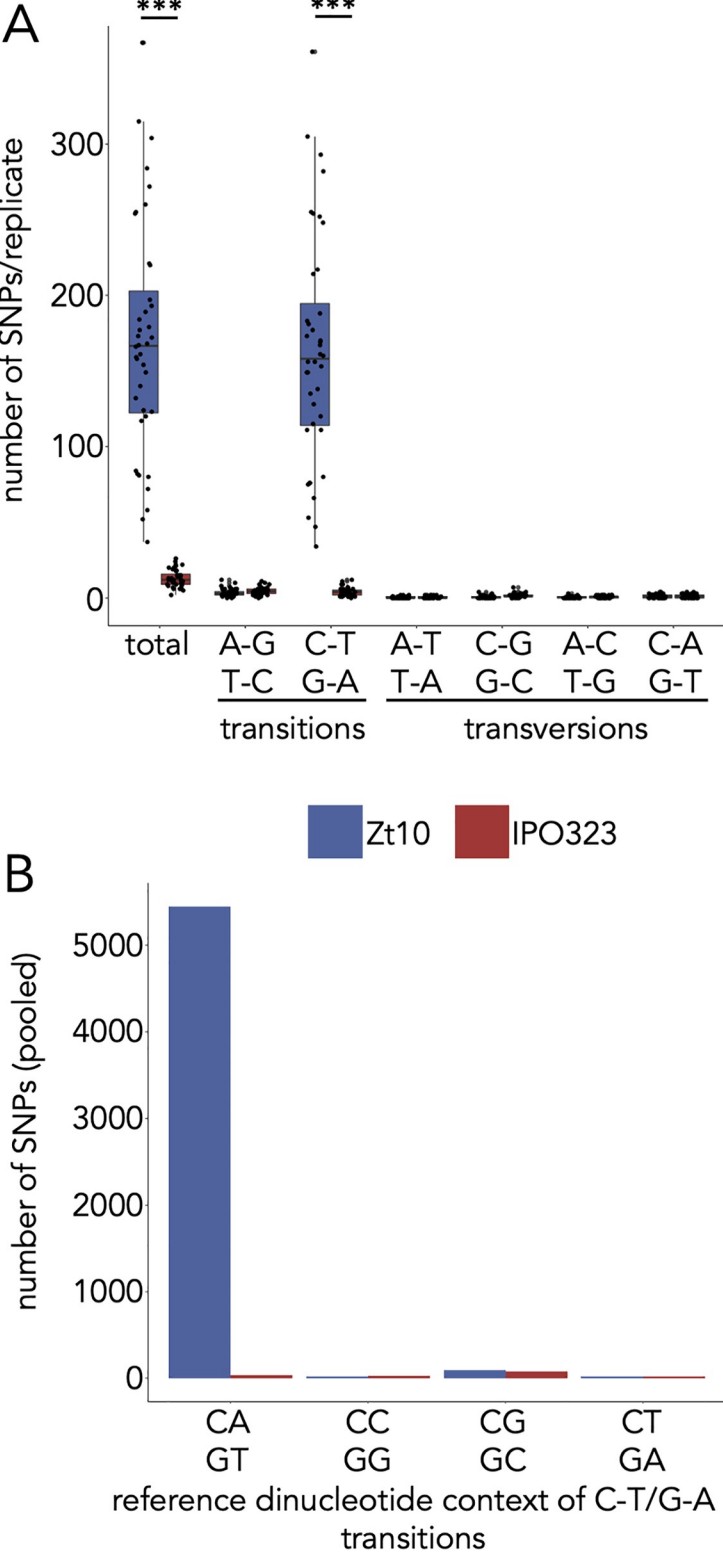

**Fig 5. Number and site preferences of SNPs in isolates with functional (Zt10) and non-functional (IPO323\*) *dim2* after one year of mitotic growth.** (A) Detected number of SNPs in the 40 replicates of each isolate. The number of SNPs in Zt10 replicates is significantly higher compared to IPO323 (Wilcoxon rank sum test, \*\*\**p*-value < 0.001). The vast majority of mutations are C to T transitions. (B) Sequence context of the mutations in the two tested isolates. The SNPs of all replicates were pooled for this analysis. C→T transitions are predominantly found in a CA context resulting in CA to TA (or GT to TA) mutations. CG sites, although preferentially methylated, are not mutated at high rates. \*Isolate Zt09 is a derivate of IPO323.

sequence repeats during mitosis. Such a mutator resembles the hitherto undescribed but much looked-for mitotic version of RIP.

## Discussion

Our study of the DNA methyltransferase *dim2* in the fungal pathogen *Z. tritici* reveals an exceptional extent of intra species polymorphism that has functional and genetic implications. Although it has been reported that 5mC is absent in *Z. tritici* [25], we found that some isolates contain a functional Dim2 *de novo* 5mC methyltransferase. We present evidence that loss of most 5mC is quite recent; we found that 5mC is not completely absent in *Z. tritici*, even when there is no functional Dim2; and we discovered a Dim2-dependent mutator phenomenon. The majority of isolates in which we identified non-mutated *dim2* genes originated in Iran, previously shown to be the center of origin of *Z. tritici* [36]. The closely related sister species *Z. ardabiliae* and *Z. brevis* were collected in Iran as well and are considered to be endemic to this region [40,49]. We found only one *Z. tritici* isolate (Zt469) that did not carry any additional *dim2* copies. This isolate was collected from *Aegilops* sp., also in Iran. We propose that the original state of the native *dim2* locus has been maintained in the two sister species and *Z. tritici* isolates collected from wild grasses. Even functional *dim2* alleles show sequence polymorphisms, i.e., transitions and in-frame deletions (between <10–100 aa in non-conserved regions), respectively. Thus, all available data suggest that *dim2* gene amplification, recombination events, and RIP have been specific to wheat-infecting *Z. tritici* isolates.

Here, we found contradicting evidence to a previous hypothesis, suggesting that the mutated *dim2* copies in IPO323 and similar strains arose by gene duplications of the native *dim2* followed by transposon-mediated movement to subtelomeric regions and RIP [25]. Instead, non-functional subtelomeric *dim2* copies show patterns of increased transversion mutations specifically in the region coding for the DNMT domain. Such patterns are not an expected outcome of RIP. Instead, we consider interspecific hybridization and introgression between distinct lineages with different *dim2* alleles. Our findings also suggest that there are at least three groups of functional *dim2* alleles at the native locus in the extant *Z. tritici* population. One group is derived from or similar to the Zt469 allele, a second group of alleles contains a 5' deletion and a third group is enriched with "transversions" when compared to the Zt469 allele. Two strains, OregS90 and CRI10, had non-functional "transversion alleles" at the native locus and we propose that both may be derived by secondary integration events. Isolate UR95 contains numerous non-functional copies but also two more recently mutated alleles, one 5' deletion and one transversion allele, both non-functional. This indicates that intact and thus presumably functional copies of these alleles are present in *Zymoseptoria* populations. The current set of genomes did not reveal a functional "transversion allele" at the native locus or subtelomeric region, but we identified a putative functional copy in WAI329, an Australian Illumina-sequenced isolate [50]. A likely scenario for repeated invasion and inactivation cycles predicts that some wild grass infecting isolates carried "transversion alleles" that were not mutated. Gene amplification or TE-mediated recombination coupled to RIP resulted in numerous non-functional alleles. Our data suggest recombination between isolates with transversion alleles in ectopic loci, which may have induced RIP of the native *dim2* allele because of high sequence similarity outside of the DNMT catalytic domain, thus resulting in non-functional *dim2* alleles only, and therefore absence of *de novo* DNMT activity. This is consistent with the previously reported signatures of re-current introgression in *Z. tritici* [31]. Further analyses of diverse *Z. tritici* isolates from different hosts and geographical locations (*T. aestivum*, *Triticum durum*, or *Aegilops* sp.) are needed to track the evolutionary history of the *dim2* gene in the *Zymoseptoria* species complex.

The previous study on *dim2* suggested absence of 5mC based on liquid chromatography coupled with electrospray ionization tandem mass spectrometry (ESI–MS/MS) [25]. This finding is consistent with the low levels of methylation we discovered in strains lacking Dim2 activity. Based on recent results obtained in studies with *Cryptococcus* we hypothesize that this low level of 5mC has been maintained by the single homologue of Dnmt5 [20,21] after the loss of Dim2. This DNMT acts as a maintenance DNMT but does not catalyze *de novo* DNA methylation [21]. In the absence of a *de novo* DNMT, DNA methylation is likely to decrease over time as any loss of 5mC cannot be restored. This idea is supported by our finding that 5mC levels are higher upon deletion of *dim2* (Zt10Δ*dim2*) compared to Zt09, where *dim2* was likely inactivated thousands of generations ago.

Sequence repeats are the predominant targets of DNA methylation in fungi [15,22,51,52], and 5mC enrichment is correlated with genome defense mechanisms that act to silence transposons [1]. We found that nucleotide composition in TEs differs between isolates and species that have functional or non-functional *dim2* alleles. CA sites, the preferred target sites of Dim2 in addition to CGs, are reduced in frequency when Dim2 is present. TA sites, however, are more frequent when Dim2 is present. This links increased C→T transition frequency to the presence of functional Dim2. Spontaneous deamination of 5mC also yields C→T transitions [48], giving rise to accelerated mutation rates at 5mC sites, a well-known phenomenon [53]. Correlation of decreased C:G to T:G transitions in absence of the DNA methyltransferase *CMT3* was recently shown in plants [54]. However, the observed differences in dinucleotide frequencies in *Zymoseptoria* and the high rate of C→T transitions in the evolution experiment suggest a role of Dim2 and 5mC in the generation of mutations. The mutations occurring throughout the evolution experiment specifically affected CAs but not CGs. If mutations resulted from spontaneous deamination, we would have expected a greater effect on CGs because they have higher 5mC levels. Lack of Dim2 resulted in complete loss of 5mC at CAs and therefore reduced CA to TA mutation frequencies. We also did not observe a noticeable difference of CG site abundance between isolates with or without functional Dim2, suggesting that non-CG 5mC (mediated by Dim2) is the main driver of the C→T transitions we observed. This suggests that, although repetitive regions are a shared target for DNMTs in different fungal species, distinct target sites of 5mC may correlate with the sequence composition of repetitive regions and the presence of other proteins involved in the 5mC pathway.

What is the underlying mechanism for the reduced CA to TA mutation frequency in absence of functional *dim2*? The best-known process causing C→T transitions in fungi is RIP, which occurs during pre-meiosis after fertilization but before karyogamy [38,39]. While RIP depends largely on the presence of the putative DNMT, RID [17], DIM-2 has also been suggested to be involved in RIP in *N. crassa* [19]. Thus, we propose that the absence of Dim2 may also play a role in the efficiency and rate of RIP in *Z. tritici*. Mechanistic studies are still lacking to address this question because crosses with engineered strains are difficult to carry out.

There is little precedence for a mutator process that generates CA to TA mutations and acts during vegetative growth in any eukaryote, the type of mutator uncovered by our evolution experiment. In *N. crassa*, RIP acts preferentially on CAs [19,39,55], and DIM-2 preferentially methylates CTs [22]. Presence of *Z. tritici* Dim2 yields non-CpG 5mC preferentially at CAs and increases C→T mutations. Essentially, the *dim2*-dependent phenomenon described here resembles what is expected for a mitotic version of RIP. We do not know yet whether Dim2 is directly involved in this mechanism and whether Rid may also play a role. However, functional alleles of *rid* are present in all strains we analyzed in the evolution experiment, and in all species examined so far *rid* is expressed solely during the sexual cycle [17]. Experiments with bacterial DNMTs uncovered inherent mutator activity under appropriate environmental conditions (e.g., low concentrations of the methyl donor, *S*-adenosylmethionine [SAM], or

AdoMet) or when specific point mutations (e.g., in the SAM-binding motif) were generated [56–61]. Another possible mechanism may involve differences in DNA repair efficiency, so that CG sites are more efficiently repaired compared to CA sites. Hemi-methylation of mutated CG sites could hereby aid in strand recognition and mismatch repair as suggested for mammalian cells [62]. Our ongoing studies are directed towards uncovering functional sequence motifs in Dim2 and examining growth conditions that increase or decrease the novel mutator phenotype.

Why are presence of widespread 5mC and differences in mutation rates polymorphic traits in this fungal pathogen? In several fungal species, 5mC is completely absent or methylation levels are very low [22,63–65] raising the question about the importance of 5mC in terms of genome defense and evolution. While silencing or inactivation of TEs protects genome integrity and stability, it may limit the adaptive potential in response to changing environmental conditions. Transposon-mediated genome diversity, gene or even chromosome copy number variations are frequently found in fungal pathogens, and they form a crucial aspect of rapid evolution [66–69]. Here we uncovered a system that suggests continuing loss and re-acquisition of the ability to silence or inactivate TEs in the genomes of numerous isolates of an important plant pathogen, *Z. tritici*. The finding that *dim2* is absent in most isolates outside the center of origin suggests that loss of *dim2* is adaptive for survival in agricultural environments. Nevertheless, in the center of origin, functional *dim2* copies are re-acquired and maintained, suggesting that 5mC is important under certain environmental conditions. The efficiency and abundance of DNA methylation and mutations may therefore represent an evolutionary trade-off between genome integrity and adaptive potential.

## Materials and methods

### Fungal isolates and growth conditions

All *Zymoseptoria* spp. isolates used in this study (S1 Table) were cultivated at 18˚C in YMS (4 g yeast extract, 4 g malt, 4 g sucrose per 1 L, 20 g agar per L for plates) medium. Cultures for DNA extraction and plant infection experiments were inoculated directly from the -80˚C glycerol stocks and grown in liquid YMS medium at 200 rpm for 5 days (DNA extractions) and in pre-cultures (3 days) and main cultures (2 days) for plant infections.

### SMRT sequencing and assembly of Iranian isolates Zt289 and Zt469

High molecular weight DNA was extracted as previously described [70]. Library preparation and PacBio sequencing was performed at the Max Planck Genome Center in Cologne, Germany (https://mpgc.mpipz.mpg.de/home/) on a Pacific Biosciences Sequel II. One SMRT cell was sequenced per genome. Genome assemblies were performed as described [71]. In brief, genomes were assembled *de novo* using the SMRT Analysis software v.5 (Pacific Bioscience) using default and "fungal" parameters. The assembly quality was determined using Quast [72] and the assembly with the best quality used for further analyses. Genome assembly statistics are summarized in S7 Table.

### Identification of TEs and analysis of TE expression

We annotated transposable elements with the REPET pipeline (https://urgi.versailles.inra.fr/Tools/REPET) [73] as described in [71]. Briefly, we identified repetitive element consensus sequences in each genome using TEdenovo following the developer's recommendations [73]. We used each library of consensus sequences to annotate genomes using TEannot with default parameters. To evaluate TE activity we analyzed expression *in planta* using RNA-seq data of *Z.*

*tritici* isolates Zt10 and Zt09 [35]. First, raw sequencing reads were quality filtered and trimmed using Trimmomatic (v0.36) [74] and mapped to the respective genome using hisat2 (v2.2.0) [75]. Reads were trimmed using the following parameters: LEADING:30 SLIDING-WINDOW:4:30 AVGQUAL:30 MINLEN:50. After mapping, the resulting aligned read files were used to assess read counts for each stage and replicate (with a total of two replicates per stage). The read count table was generated using TEcount function of the TEtranscript pipeline with the following parameter: -mode multi [76]. Briefly, TEtranscript pipeline counts both uniquely and multi-mapped reads mapped to annotated transposons and genes to attribute transcript abundance. A given TE was considered transcribed only if reads mapped all along the element length, in order to avoid redundancy [76]. Levels of expression were calculated using Transcript per Million (TPM) normalization which corresponds to the normalization of read counts with transposon (or gene) length per million. We considered a TE as 'expressed' if $TPM > 0$.

## Sequence identification and comparison of DNA methyltransferases

All analyzed genomes and isolates used in this study are listed in S1 Table and originate from the following studies [31–36,40,49,71,77–82]. We identified homologs of *dim2* using the predicted 'deRIPed' protein sequence of *Z. tritici* IPO323 [25] as a template. To compare the sequence identity between active and inactive *dim2* copies in the genome, we used *dim2* of isolate Zt469 as a reference and performed pairwise comparisons. To identify additional putative DNA methyltransferases, we used the DNA methyltransferase domain of the Dim2 protein as query for a BLAST search against the *Z. tritici* genomes. BLAST searches, phylogenetic trees and alignments were performed using Geneious 'Blast', Geneious 'Alignment' and Geneious 'Map to reference' (Geneious version 10.2.4 (http://www.geneious.com, [83]). A distance tree based on the nucleotide sequence of *dim2*, *rid* and *dnmt5* was generated with the following settings: alignment type: global alignment with free end gaps, cost matrix: 65% similarity, genetic distance: Jukes-Cantor, tree build method: neighbor joining, outgroup: *Zymoseptoria passerinii* (Zpa63).

## Generation of *dim2* deletion and integration strains

We transformed the *Z. tritici* isolates Zt09 and Zt10 using an *Agrobacterium tumefaciens*-mediated transformation (ATMT) protocol as previously described [84]. Briefly, we created plasmids containing the integration (pES189) or deletion (pES188) constructs using Gibson assembly [85] (S8 Table). Both constructs contained the hygromycin resistance cassette as selection marker. Plasmids were amplified in *E. coli* TOP10 cells and sequenced to confirm correct assembly of the constructs followed by electroporation of *A. tumefaciens* strain AGL1. *Zymoseptoria tritici* strains were transformed with the respective *A. tumefaciens* strains by co-incubation for 3–4 days at 18˚C on induction medium. Following co-incubation, the strains were grown on selection medium containing hygromycin to select for integration of the constructs and cefotaxime to eliminate *A. tumefaciens*. Single *Z. tritici* colonies were selected, streaked out twice and the correct integration of the construct was verified by PCR and by Southern blot analyses (S4 Fig).

## Bisulfite treatment and sequencing

We extracted high molecular weight DNA of three biological replicates of Zt10 and Zt09 and three independent transformants of Zt10Δ*dim2* and Zt09::*dim2* as described previously [70]. Genomic DNA was sent to the Max Planck Genome Centre in Cologne, Germany (https://mpgc.mpipz.mpg.de/home/) for bisulfite treatment and sequencing. Lambda DNA was used

as spike-in (~1%) to determine conversion efficiency. Genomic DNA was fragmented with a COVARIS S2 and an Illumina-compatible library was prepared with the NEXTflex Bisulfite Library Prep Kit for Illumina Sequencing (Bioo Scientific/PerkinElmer, Austin, TX). Bisulfite conversion was performed using the EZ DNA Methylation Gold Kit (Zymo Research, Irvine, CA). Illumina sequencing was performed on a HiSeq3000 machine with paired-end 150-nt read mode (S9 Table).

To confirm bisulfite sequencing results, we treated genomic DNA of the fungal strains and, as a control for conversion efficiency, the Universal Methylated DNA Standard (Zymo Research) (~100 ng) with bisulfite using the EZ DNA Methylation-Lightning Kit (Zymo Research) according to manufacturer's instructions. Two representative loci per fungal strain and the human MLH1 for the control were amplified by PCR (ZymoTaq PreMix, Zymo Research) using specifically designed primers for amplification of the bisulfite-treated DNA (designed using the 'Bisulfite Primer Seeker' (Zymo Research), S8 Table). PCR products were cloned using the TOPO TA kit (Thermo Fisher Scientific) and Sanger sequenced (Eurofins Genomics, Ebersberg, Germany). Analysis of sequenced bisulfite-converted DNA sequences was performed using Geneious software version 10.2.4 [83].

## Data analysis of whole genome bisulfite sequencing data

A list of software and input commands used in our analyses is provided in the S1 Text.

Reads were quality filtered using Trimmomatic (v0.36) [74] and subsequently mapped using Bismark (v0.20.0) [86]. Duplicate reads were removed. Methylated sites were extracted using the bismark_methylation_extractor applying the–no_overlap and CX options. We considered sites methylated, if at least 4 reads and $\geq$ 50% of reads supported methylation. We then extracted these sites (see S1 Text) in CG, CHG or CHH contexts for further analysis. Bedtools (v2.25.0) [87] was used to correlate methylation and genomics features.

## Southern blots to detect DNA methylation and confirm *dim2* mutant strains

To detect presence or absence of 5mC, we performed Southern blots according to standard protocols [88]. Genomic DNA was extracted using a standard phenol-chloroform extraction method [89]. The same amount of DNA and enzymes (*Bfu*CI, *Dpn*I, *Dpn*II; 25 units, New England Biolabs, Frankfurt, Germany) was used as input for the different restriction digests to make restriction patterns comparable between enzymes and *Z. tritici* isolates for the detection of DNA methylation. Probes were generated with the PCR DIG labeling Mix (Roche, Mannheim, Germany) following the manufacturer's instructions and chemiluminescent signals were detected using the GelDoc XR+ system (Bio-Rad, Munich, Germany).

## Phenotypic assay *in vitro*

Spores were diluted in water ($10^7$ cells/mL and tenfold dilution series to 1,000 cells/mL) and three μL of the spore suspension dilutions were pipetted on plates and incubated for ten days. To test for responses to different stress conditions *in vitro*, YMS plates containing NaCl (0.5 M and 1 M), sorbitol (1 M and 1.5 M), Congo Red (300 μg/mL and 500 μg/mL), $H_2O_2$ (1.5 mM and 2 mM), methyl methanesulfonate (0.01% and 0.005%) and two plates containing only YMS were prepared. All plates were incubated at 18˚C, except for one of the YMS plates that was incubated at 28˚C.

## Phenotypic assay on wheat

Seedlings of the wheat cultivar Obelisk (Wiersum Plantbreeding BV, Winschoten, The Netherlands) were pre-germinated on wet sterile Whatman paper for four days under normal growth

conditions (16 h at light intensity of ~200 μmol/m$^{-2}$s$^{-1}$ and 8 h darkness in growth chambers at 20˚C with 90% humidity) followed by potting and growth for an additional seven days. Marked areas on the second leaves (30 leaves per strain) were inoculated with a spore suspension of 10$^7$ cells/mL in H$_2$O and 0.1% Tween 20. Mock controls were treated with H$_2$O and 0.1% Tween 20 only. 23 days post inoculation, treated leaves were analyzed for infection symptoms in form of necrosis and pycnidia. Evaluation was performed by assigning categories for necrosis and pycnidia coverage to each leaf (categories: 0 = 0%, 1 = 1–20%, 2 = 21–40%, 3 = 41–60%, 4 = 61–80%, 5 = 81–100%).

## Gene expression analysis

Expression of *dnmt5* and *rid* in Zt09 and Zt10 as well as of *dim2* in Zt10 was analyzed using previously generated *in planta* and *in vitro* RNA-seq data [35,41]. *In planta* data refers to the previously characterized morphological wheat infection stages: (A) infection establishment, (B) biotrophic growth, (C) lifestyle transition, (D) necrotrophic growth and reproduction [35]. *In vitro* data reflects growth of Zt09 in liquid YMS medium at 18˚C and 200 rpm [41]. In brief, reads were quality filtered and trimmed as described in [35] and mapped to the respective *Z. tritici* genomes [37,71] using hisat2 v2.2.0 [90]. Relative abundance of transcripts for all Zt09 and Zt10 genes [46,71] was calculated in RPKM using Cuffdiff2 v2.1.1 [91].

## Analysis of dinucleotide frequencies

TE sequences were extracted from the genome using bedtools getfasta [87]. k-mer frequencies in annotated TEs and masked genomes were determined using the software jellyfish (version 2.3.0) [92]. Correspondence analyses were carried out and visualized in R [93] using the packages "FactoMineR" (v2.3) and "factoextra" (v1.0.7) [94].

## Mutation accumulation experiment and SNP analysis

A single colony derived directly from a plated dilution of frozen stock for Zt10 or IPO323 was resuspended in 1 mL YMS including 25% glycerol by 2 min vortexing on a VXR basic Vibrax at 2000 rpm, and 10–50 μL were re-plated onto a YMS agar plate. Forty replicates were produced. Cells were grown for 7 days at 18˚C until a random colony (based on vicinity to a prefixed position on the plate) derived from a single cell was picked and transferred to a new plate as described above. The transfers were conducted for one year (52 times) before the DNA of a randomly chosen colony of each replicate was extracted and sequenced. Sequencing and library preparation were performed at the Max Planck Genome Center in Cologne, Germany (https://mpgc.mpipz. mpg.de/home/). Sequencing was performed on an Illumina HiSeq2500 machine obtaining paired-end 250-nt reads (S9 Table). Paired-end reads were quality filtered (Trimmomatic v0.38), mapped (bowtie2 v2.3.5) and SNPs were called using samtools (v1.7) mpileup (see S1 Text).

## Supporting information

**S1 Fig. Phylogenetic trees of DNA sequences encoding the *dim2*, *dnmt5*, and *rid* genes. (A)** Phylogenetic tree based on alignments of "deRIPed" *dim2* alleles to the native, functional gene of Zt469. The *Z. tritici dim2* is distinct from the *Z. brevis* and *Z. ardabiliae* gene. The native *Z. tritici* copies, except for OregS90 and CRI10, and non-native copies form distinct clusters. Shown are two representative "deRIPed" non-native copies per isolate. **(B)** *Dnmt5* is present in all analyzed genomes and shows relatively little inter- or intraspecies diversity. **(C)** The *rid gene* shows an exceptionally high inter- and intraspecies diversity with three highly distinct alleles present among genomes of *Z. tritici*, *Z. ardabiliae* and *Z. brevis*. Green background

indicates *Z. tritici*, blue *Z. ardabiliae*, yellow *Z. brevis*. *Z. passerinii* is used as an outgroup.
(PDF)

**S2 Fig. Alignment of original (A) and 'deRIPed' (B) full-length (> 3000 bp) *dim2* alleles compared to the functional allele of isolate Zt469.** All native copies, (functional and non-functional, except for OregS90 and CRI10) lack mutations in the DNA methyltransferase domain (position ~2,300–3,500) that are present in all non-native copies suggesting that the additional copies did not emerge from amplification of the native *dim2*. Black lines indicate differences to the reference.
(PDF)

**S3 Fig. Protein alignment of eight known or presumed functional and two de-RIPed, non-functional alleles. (A)** Cartoon of complete protein coding sequences. The conserved domains detected by homology searches are indicated above the alignment (BAH, Bromo-Adjacent Homology domain). The top four alleles (Zt467, Zt468, Zt469, Zt471) were identified in strains isolated from *Aegilops*, all others were isolated from wheat. Alleles from CRI10 and Zt09 are non-functional and were "de-RIPed" as described in the main text. Sequences of alleles from Zt289 and TN09 suggest the presence of several potentially functional *dim2* alleles in natural populations of *Z. tritici*. Color scheme according to rasmol (http://www.pdg.cnb.uam.es/cursos/Barcelona2002/pages/Farmac/CMU_course/ProtG/ShapelyColors.html). **(B)** Complete sequence alignment of all alleles shown in the cartoon. All SNVs and indels that lead to non-synonymous changes are indicated; there are many more synonymous than non-synonymous mutations in the various alleles. The conserved ten canonical motifs found in most DNMTases are underlined and indicated with Roman numerals. Motif I is the SAM (Adomet)-binding domain, motif IV includes the catalytic site with the invariant Pro-Cys site. Motif VIII is the most variable in *Zymoseptoria dim2* alleles–this region is considered a poorly conserved targeting domain in bacterial DNMTases. Amino acids indicated in bold red show variations likely caused by the deRIP procedure (i.e., C or G were introduced when T or A would have retained the conserved amino acid, e.g. TCN->CCN, Ser, conserved, to Pro, de-RIP, or CAR->CGR, Gln, conserved, to Arg, de-RIP). Amino acids shown in blue suggest the presence of two or more different alleles in the *Z. tritici* population; most of these differences occur at the non-conserved C-terminus. Mutation of the predominant TAG/A stop codon to CAG/A (Q) generated a short addition of nine amino acids in two alleles. In addition to the known Dim2 domains, the two BAH (aa 494–601 and 637–748) and DNMT catalytic domain (aa 694–1171), we predict several other motifs that are conserved in Dim2 proteins from a wide variety of fungi, covering all groups of ascomycetes. This includes unexpected weak similarity to a domain that may be important to the targeting of DNMT1, the mammalian maintenance DNMT, to replication foci (RFD, replication focus domain), another short motif (TIVTPF) that is present in almost all Dim2 proteins but embedded in variable sequences, a putative nuclear localization signal (NLS), and two short motifs that are nearly invariable in Dim2 proteins (FPDHR and PTRPHGVGL). The only variant amino acid that may affect function is a D->A change predicted in the deRIPed version of the non-functional Zt09 *dim2* allele at the end of the second BAH domain (orange A). All other differences are predicted to have only minor, if any, consequences for Dim2 activity.
(PDF)

**S4 Fig. Southern blots to confirm the correct integration of (A) the deletion construct for *dim2* in Zt10 (Zt10Δ*dim2*) and (B) *dim2* (originating from Zt10) in Zt09 (Zt09::*dim2*).** Three positive transformants (#49, #63 and #84) were found amongst the Zt09::*dim2* candidates, whereas all seven candidates for Zt10Δ*dim2* were verified (correct transformants are

highlighted in bold).
(PDF)

**S5 Fig. Phenotypic characterization of Zt10, Zt10Δ*dim2*, Zt09 and Zt09::*dim2 in vitro*.** We compared growth phenotypes under different *in vitro* conditions including temperature, osmotic, oxidative, genotoxic and cell wall stress. We spotted spore dilutions of each reference isolate and three independent *dim2* mutant transformants on each plate. We did not detect any noticeable differences in growth between reference and mutant strains. Previously described differences between the *Z. tritici* isolates Zt09 and Zt10 were observed [35].
(PDF)

**S6 Fig. Results of wheat infection experiments with *Z. tritici* isolates Zt09, Zt10 and the mutants Zt09::*dim2* and Zt10Δ*dim2* on wheat. (A)** Zt09::*dim2* strains show significantly less necrotic lesions compared to Zt09 (*** Wilcoxon rank-sum test, *p*-value = 2.178 x $10^{-7}$) while there is no significant difference in the quantities of necrotic lesions caused by Zt10 and Zt10Δ*dim2* strains. **(B)** Coverage with pycnidia is significantly reduced between Zt09 and Zt09::*dim2* (** Wilcoxon rank-sum test, *p*-value = 0.003301) but not between Zt10 and Zt10Δ*dim2* strains. Categories for necrotic lesion and pycnidia coverage: 0 = 0%, 1 = 1–20%, 2 = 21–40%, 3 = 41–60%, 4 = 61–80%, 5 = 81–100%.
(PDF)

**S7 Fig. Detection of cytosine methylation by enzymatic digestion and Southern blot and confirmation of bisulfite sequencing results by PCR and Sanger sequencing.** Restriction enzyme analysis followed by Southern blots using the rDNA spacer **(A)** or a retrotransposon RIL2 **(B)** as probe. In addition to the cytosine methylation sensitive (*Bfu*CI, +) or insensitive (*Dpn*II, -) enzymes, we used *Dpn*I (*) to test for the presence of adenine methylation. Zt09 and Zt150 contain a non-functional *dim2*, whereas the Iranian strains Zt10, Zt11 and the *Z. ardabiliae* strain Za17 have a functional copy. In all strains with a functional *dim2* we see a clear difference between the restrictions with *Bfu*CI and *Dpn*II, indicating the presence of DNA methylation. In Zt09 and Zt150 this difference is not detectable except for one band that is not restricted in both strains in the rDNA blot. The genomic DNA of Zt09 treated with *Dpn*I is not digested suggesting absence of 6mA methylation in these regions. **(C)** Confirmation of bisulfite sequencing by an independent bisulfite treatment followed by PCR of target loci and Sanger sequencing. Two target loci (repeated region that showed 5mC signals in all isolates) were chosen per isolate, shown are the number of detected 5mC sites in these loci (size of loci ~ 180–260 bp). N is the number of cloned PCR products that were sequenced. Based on these data we can confirm the presence of 5mC methylation in all isolates and a lower frequency in absence of a functional *dim2*.
(TIF)

**S8 Fig. 5mC site preferences in CG, CHG and CHH sequence contexts. (A)** CGA and CGC sites are preferred in CG contexts in presence of *dim2*, while CGA and CGG are slightly preferred targets in absence of *dim2*. **(B)** Among CHG sites, CAG sites are the predominant target sites of 5mC. For CHG and CHH contexts only data for Zt10 and Zt09::*dim2* are shown, as there is no detectable methylation outside of CG contexts in absence of a functional *dim2*. **(C)** In CHH contexts, CA sites followed by CC sites have the highest methylation frequency. CT sites are almost completely devoid of 5mC.
(PDF)

**S9 Fig. Methylation levels (A) and overlap of 5mC sites between the three replicates (B).**
(A) Methylation levels of 5mC sites (5mC reads/site) in the different mutants and replicates.

Methylation levels do not drop below 50% as this was the threshold to call 5mC sites. Methylation levels are highest in Zt09, where only few CpG sites are maintained by Dnmt5. All other strains have methylation levels mostly between 70–80% indicating more heterogenity of 5mC sites within the cell populations. (B) Overlap of detected 5mC sites between replicates of the same strain. Zt10 and Zt09 replicates originate from three independent inoculations while the mutant replicates represent independent transformants. The largest differences in terms of 5mC site overlap can be observed in the Zt10Δ*dim2* mutant. A likely explanation is that 5mC sites are in the process of being lost in absence of *dim2* and that different sites get lost at different times in the independent transformants.
(PDF)

**S10 Fig. Expression pattern of the three (putative) DNA methyltransferases *dim2*, *dnmt5* and *rid* during plant infection and *in vitro* in the isolates Zt10 and Zt09.** *In vitro* data is only available for Zt09. Expression during infection was monitored in specific stages: A–infection establishment, B–biotrophic growth, C–transition from biotrophic to necrotrophic growth, D–necrotrophic colonization [35].
(PDF)

**S11 Fig. Predicted domains of the three putative DNA methyltransferase proteins Dim2, Dnmt5 and Rid identified in the *Zymoseptoria* genomes.** Protein sequences were analyzed with SMART [95], NCBI Blast and InterProScan [96].
(PDF)

**S1 Table. List of *Z. tritici*, *Z. ardabiliae*, *Z. brevis* and *Z. passerinii* isolates used in this study.** Listed are collection date, origin, information about the genome assembly and presence/absence of *dim2* in each isolate.
(XLSX)

**S2 Table. Detected copies and pairwise identity of full-length (>3,000 bp) *dim2* copies in *Z. tritici* genomes.** The *dim2* gene of the Iranian isolate Zt469 was used as query for the blast search.
(XLSX)

**S3 Table. Dinucleotide frequencies in transposable elements (TEs) and TE masked genomes in *Z. tritici* isolates and the sister species *Z. brevis*, *Z. ardabiliae* and *Z. passerinii*.**
(XLSX)

**S4 Table. Genes overlapping TEs and their functional annotation in Zt09.**
(XLSX)

**S5 Table. SNPs detected in Zt10 and IPO323 after 52 weeks of experimental evolution compared to the reference strain at the start of the experiment.**
(XLSX)

**S6 Table. Genome-wide occurrence of 5mC and SNPs in Zt10.** Listed are GC and AT content, number of 5mC sites (pooled data from WGBS of all three replicates) and number of SNPs (pooled data from all 40 replicates) in 500 bp windows.
(XLSX)

**S7 Table. PacBio genome assembly metrics of Iranian isolates Zt289 and Zt469.**
(XLSX)

**S8 Table. List of all oligos and plasmids used in this study.**
(XLSX)

**S9 Table. Overview of Illumina sequencing data, detected 5mC sites, and SNPs.**
(XLSX)

**S1 Text. Software and commands used for bisulfite and genome data analysis to detect 5mC sites and SNPs.**
(DOCX)

## Acknowledgments

We thank Bruce A. McDonald for providing Iranian *Z. tritici* isolates, Fatemeh Salimi for sampling Iranian *Z. tritici* isolates, Kathrin Happ, Anja Lachner, Maja Stralucke, Kim Hufnagel and Doreen Landermann for assistance with experiments and John B. Ridenour for comments on the manuscript. EHS is a CIFAR fellow and acknowledges support.

## Author Contributions

**Conceptualization:** Mareike Möller, Michael Habig, Michael Freitag, Eva H. Stukenbrock.

**Formal analysis:** Mareike Möller, Michael Habig, Cécile Lorrain, Alice Feurtey, Michael Freitag, Eva H. Stukenbrock.

**Funding acquisition:** Mareike Möller, Michael Freitag, Eva H. Stukenbrock.

**Investigation:** Mareike Möller, Michael Habig, Cécile Lorrain, Alice Feurtey, Janine Haueisen, Wagner C. Fagundes, Michael Freitag.

**Project administration:** Eva H. Stukenbrock.

**Resources:** Janine Haueisen, Wagner C. Fagundes, Alireza Alizadeh.

**Supervision:** Michael Freitag, Eva H. Stukenbrock.

**Visualization:** Mareike Möller.

**Writing – original draft:** Mareike Möller, Michael Freitag, Eva H. Stukenbrock.

**Writing – review & editing:** Mareike Möller, Michael Habig, Cécile Lorrain, Alice Feurtey, Janine Haueisen, Wagner C. Fagundes, Michael Freitag, Eva H. Stukenbrock.

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
