## [Decision Letter · Decision Letter 0]

27 Oct 2020

Dear Dr Möller,

Thank you very much for submitting your Research Article entitled 'Recent loss of the Dim2 DNA methyltransferase decreases mutation rate in repeats and changes evolutionary trajectory in a fungal

pathogen' to PLOS Genetics. Your manuscript was fully evaluated at the editorial level and by independent peer reviewers. The reviewers appreciated the attention to an important problem, but raised some substantial concerns about the current manuscript. Based on the reviews, we will not be able to accept this version of the manuscript, but we would be willing to review again a much-revised version. We cannot, of course, promise publication at that time.

If you decide to revise the manuscript for further consideration at PLOS Genetics, please aim to resubmit within the next 60 days, unless it will take extra time to address the concerns of the reviewers, in which case we would appreciate an expected resubmission date by email to plosgenetics@plos.org.

[LINK]

We are sorry that we cannot be more positive about your manuscript at this stage. Please do not hesitate to contact us if you have any concerns or questions.

Yours sincerely,

Ksenia Krasileva

Guest Editor

PLOS Genetics

Kirsten Bomblies

Section Editor: Evolution

PLOS Genetics

The manuscript received most enthusiastic and supportive feedback from all reviewers all of whom unanimously agreed that the study presents major scientific advancement in area of fungal genome evolution and DNA methylation. We invite you to revise your manuscript addressing all of the reviewer’s comments and provide it for re-review. All reviewers agreed that experimental evolution is at the center of this study. Please, consider and address carefully the critique of experimental evolution section by the first reviewer providing a more in-depth explanation of the choice of strain used in experimental evolution. In this context, it is important discuss in more detail other differences besides dim2 between the Zt09 and Zt10 strains. Two of the reviewers comment on the origin and functionality of dim2 alleles that needs careful consideration. The third reviewer raises an important issue of interpreting Zt09:DIM2 virulence phenotypes which also need closer attention. Finally, provide more details on methodology and conclusions used throughout the study as outlined by reviewers.

Reviewer's Responses to Questions

**Comments to the Authors:**

Reviewer #1: The manuscript 'Recent loss of the Dim2 DNA methyltransferase decreases mutation rate in repeats and changes evolutionary trajectory in a fungal pathogen' by Möller and colleagues reports on computational and experimental analyses of the Dim2 DNA methyltransfereses in the fungal wheat pathogen Zymoseptoria tritici. The authors demonstrate that, next to conserved rid and dnmt5, few Z. tritici strains isolated at the centre of origin carry intact dim2 while many other isolates encode (multiple) inactivated versions of this gene. The authors furthermore show that dim2 is responsible for de novo DNA methylation (5mC) at transposable elements (TEs). Intriguingly, by performing experimental evolution experiments, the authors observed increased nucleotide changes C->T on CpA sites during mitosis. These mutations could contribute to the inactivity of TEs which therefore would shape the TE repertoires in strains polymorphic in dim2. This mutational processes seems to be analogous to the RIP genome defense mechanisms, which is RID dependent and thus far has been linked to the sexual cycle of fungi. This here described dim2-dependent mutator could therefore explain the occurrence of RIP-like mutations in many fungal genomes that lack apparent sexual cycles.

I really enjoyed reading this paper. It is very clearly written, the data is presented in a very clear manner, and the results seem to support the authors' main conclusions. However, I am concerned about some aspects of the experimental evolution experiment the authors conducted to show that dim2 promotes C->T mutations during mitosis. It is unclear why the authors use a different isolate with non-functional dim2 (IPO323) compared with Zt09 that is used throughout the remainder of the manuscript. The authors indicated that Zt09 is derived from IPO32. Based on information I found in other papers from this group, Zt09 seems to be an isolate with a deletion of chromosome 18. Thus, it is at least questionable to which extend the observations made on Zt09 throughout the manuscript can be easily transferred to IPO323 as it is not clear to which extent the genotypes differer (except del chr. 18, e.g. accumulation of other mutations throughout the genome). Furthermore, Zt09/IPO323 and Zt10 have been isolated from different regions and at different time points, and it is therefore conceivable that the genotypes of Zt10 and IPO323 differ not only in dim2. Consequently, the increased number of mutations in the evolved strains compared with the reference might therefore not only be due to differences in dim2. To address these concerns, the authors need to corroborate their observations by performing the evolution experiments by using the deletion/complementation strains of Zt09 and Zt10 as here the genetic background is identical.

Some further detailed comments/suggestions:

L112: Could the authors please provide more detail how full-length dim2 was identified in Zt10?

L221: The authors observe that introduction of dim2 into a dim2 deficient strain (Zt09) reduces virulence while deletion of dim2 in a dim2 competent strain (Zt10) does not show a corresponding phenotype (increase in virulence). Furthermore, the authors suggest (L485) that loss of dim2 is adaptive in an agricultural setting due to increased virulence. The authors really need to explain this observation in the first place and why increased DNA methylation would impact virulence.

L245: Could the authors provide a bit more detailed on the distribution of 5mC in the genome. How does the distribution of methylation levels look like (number of mC reads over C per position)?

L285: Obviously, this paper focuses on dim2 rather than on dnmt5. However, I feel the authors might have missed a huge opportunity by not providing additional functional data (e.g. deletion/complementation and WGBS) on the role of dnmt5. Recent work by Bewick and colleagues demonstrated that the majority of fungi encode dim2, dnmt5, and rid (Bewick et al. 2019). However, most work to my knowledge on DNA methylation in fungi has focused on systems with only two DNA methyltransferases (dim2 and rid, e.g. Neurospora). Studying a system with all three enzymes would provide an intriguing opportunity.

L297: How do the authors explain the grouping of UR95 in the correspondence analyses?

L397: This is similar to the point raised for Fig 2. The authors seem to infer that dim2 alleles with polymorphisms except premature stop codons are functionally active. First, do this polymorphisms alter amino-acid residues, for instance in the catalytic domain. Second, to ascertain that these alleles are functional, the authors need to provide additional data that these alleles are indeed functional.

L467: Could the authors provide more details how the different DNA methlytransferases are expressed in Z. tritici? Is dim2, for example, constitutively expressed?

L503: Could the authors please provide some brief details how genome assemblies were performed?

L530: Was the search with the DNA methyltransferase domain done using BLAST or via profile search against the genome?

Fig. 2: The authors identify multiple dim2 that contain an 5' deletion. Nevertheless, they conclude that some of these might be intact, thus active. For example, Zt289 contain this deletion. However the authors do not provide any evidence that this allele is indeed functional. I would suggest to at least briefly address this possible ambiguity in the results/discussion.

Fig. 3: Can the authors please provide an explanation why the number of methylated sites is higher in the Zt09::dim2 strain compared with Zt10? is there a difference in copy-number of expression of the dim2 allele between the isolates?

Fig. 4: For clarity, could the authors please indicate in panel a and b the different dim2 classes (active, inactive, etc.) based on e.g. differences in color or shapes?

Reviewer #2: Although a wheat pathogen, Zymoseptoria tritici is a tractable lab model to study fungal genome evolution. In this manuscript, Möller et al. present the identification of Z. tritici isolates carrying a functional dim2 gene while strains of reference display inactivated dim2 genes and low levels of 5mC. They showed that presence of functional dim2 alleles, either from endogenous or transgenic origin, correlates with significant levels of 5mC in TEs. Importantly, an evolution experiment conducted over 1000 mitotic divisions reveals that functional dim2 enzyme alters nucleotide composition of TE, in a RIP independent manner. This finding is a real breakthrough in the field of fungal genome evolution. Overall, the quality of this manuscript is suitable for publication in PLOS Genetics, providing some modifications and clarifications that are listed below.

The major concern deals with the mutation patterns of the de-RIPed non-native dim2 alleles (Figure 2), which strongly suggest a clonal origin of all of the sequenced subtelomeric copies. Consequently, the authors’ statement that “all non-native alleles share a specific pattern of transition and, surprisingly, transversion mutations in the DNA methyltransferase domain” does not fully fit with these data. More than specific, these are almost identical patterns, likely inherited from a single integration event of the so-called ‘transversion allele’ in an ancestral strain lacking dim2 activity. The text should be modified adequately. The authors should also consider using the de-RIPed dim2 phylogenetic analysis present in this study (Figure S1 A) to address questions linked to the clonal origin of the ‘transversion allele’, namely when and where (i.e. in which strain) this integration event might have happened.

The “native dim2 gene” expression is misleading since “native” would mean “functional” is most readers’ mind, whereas here, it is used to qualify a genomic location rather than a functional feature. So I would rather use “resident dim2 gene” or alternatively “dim2 native locus”.

- Abstract: “Integration of a functional dim2 allele in strains with mutated dim2 restored normal 5mC levels, demonstrating de novo cytosine methylation activity of dim2 this is an overstatement”.

The authors did not demonstrate de novo methylation activity of dim2 and alternative possibilities exist. For one, 5mC patterns could very well be installed by non-DIM2 DNMT(s) and further maintained by DIM2.

- line 120: “Five Z. tritici isolates contained an intact, non-mutated copy of dim2 in addition to multiple mutated, non-functional copies”.

How is it possible that an intact non-mutated copy of dim2 persists, meaning that it escaped RIP, when multiple mutated, non-functional copies are also present in the same haploid genome? Are they some data related to RIP efficiency on repeats located in cis vs in trans in Z. tritici’s genome? Alternatively, would you postulate that these strains did not reproduce sexually?

- p148: Any polymorphism among functional Dim2 alleles of the strains from Iran ? On a more general note, it would be interesting to give more evolutionary depth to this descriptive paragraph by performing dN/dS analyses, at least for conserved transversion mutations within the catalytic domain, eventually compared to the few ones outside of it.

- line 157: Are they from a specific retrotransposon family?

- line 182: “This second pattern was unexpected; the most parsimonious explanation is replacement of non-functional alleles at the native locus with novel dim2 alleles, perhaps by recombination with wild grass infecting isolates, such as Zt469.”

On what ground? The authors should underpin this hypothesis with further mechanistic explanation. Is there any established example of such a replacement to refer to? Besides, the authors stated in the discussion section that “Further analyses of diverse Z. tritici isolates from different hosts and geographical locations (T. aestivum , Triticum durum , or Aegilops sp.) are needed to track the evolutionary history of the dim2 gene in the Zymoseptoria species complex. ” Therefore if it is not known for sure that wheat-pathogen strains and wild-grass infecting strains are present at the locations (Table S2), how to argue for inter-specific recombination events?

- Figure 2A: It would be handy to give some structural information about end/start of the dim2 CDS and location of conserved N-term motifs of the dim2 enzyme.

- line 229 and Fig S6: in order to properly demonstrated that fragments of higher molecular weight generated by 5mC-sensitive enzymes are readily due to DNA methylation and not to partial enzymatic reactions, these blots have to be hybridized using a single unmethylted gene as a probe.

- line 255: “a de novo DNA methyltransferase is present.”

Which one are you referring to?

- lines 267-286: What is the point of this paragraph? This is more of general background and should be moved either in S8 figure legend for the first part or in discussion section for the last sentences.

-lines 415-416: “A likely scenario for repeated invasion and inactivation cycles predicts that some wild grass infecting isolates carried “transversion alleles” that were not mutated.”

If the ‘tranversion alleles’ are the non-native mutated ones showed fig2A, this sentence is puzzling.

- lines 482-484: “Here we uncovered a system of continuing loss and re-acquisition of the ability to silence or inactivate TEs in the genomes of numerous isolates of an important plant pathogen, Z. tritici.”

Still speculative, this sentence should be rephrased.

Reviewer #3: I would like to congratulate the authors on a great manuscript that I already enjoyed reading the preprint. I also would like to apologize for being really late with the reviews.

The manuscript describes an evolutionary study of the DNA methyltransferase DIM2 in the important plant pathogen Z. tritici. The study is very timely and very well executed. I only have a couple of minor suggestions and comments that are easily addressed by re-writing.

Minor points:

* p. 6 140ff and Figure 2. It would be great if the text and the figure could be linked some more. Right now this is a bit difficult to follow. For example, l 179 mentions pattern 2 and this is not explained in the figure. I think the authors talk about three patterns. This should be more explicit in the text and in the figure.

* I am a bit confused by the usage of dim2 as active and inactive. The common conventions followed is that functional alleles are all capital italic "DIM2", non-functional low case italic dim2, and protein variants as all capital DIM2. This is potentially something the authors want to take on board.

* The authors state that methylation has a negative effect on infection in wheat based on Fig. S5 where a functional DIM2 is transformed into Zt09 (and potentially that Zt10 having a functional allele is not as infective). What is the sample size of transformants tested here? Could this be a pleitropic effect of the gene insertion? How does the mutation of Zt09:DIM2 alter this phenotype? There is no need to do new experiments if these are too long-winded. Yet I would encourage the authors to consider these alternative explanations for the observed reduced virulence in this transgenic Z. tritici line.

* The insight gained by the correspondence analysis wasn't immediate obvious to me. I am also not familiar with this analysis. This could be explained a bit more or supported with additional analysis e.g. what are the relative abundance of CG or GA sides in +dim2 lines TEs?

* The method section could include version numbers for all programs used with specific flags if any were evoked.

I really enjoyed the evolution experiment. Nicely done.

I don't have anything to add to the discussion as my suggestion were already included based on the comments on the preprint.

**Have all data underlying the figures and results presented in the manuscript been provided?**

Reviewer #1: Yes

Reviewer #2: Yes

Reviewer #3: None

PLOS authors have the option to publish the peer review history of their article (what does this mean?). If published, this will include your full peer review and any attached files.

Reviewer #1: No

Reviewer #2: No

Reviewer #3: **Yes: **Benjamin Schwessinger

---

## [Decision Letter · Decision Letter 1]

26 Feb 2021

Dear Dr Möller,

We are pleased to inform you that your manuscript entitled "Recent loss of the Dim2 DNA methyltransferase decreases mutation rate in repeats and changes evolutionary trajectory in a fungal

pathogen" has been editorially accepted for publication in PLOS Genetics. Congratulations!

Yours sincerely,

Ksenia Krasileva

Guest Editor

PLOS Genetics

Kirsten Bomblies

Section Editor: Evolution

PLOS Genetics

Comments from the reviewers (if applicable):

The reviewers found that their comments were addressed. Great job!

Reviewer's Responses to Questions

**Comments to the Authors:**

Reviewer #2: The authors responded to all points I raised.

Reviewer #3: I really appreciated the very thoughtful responses to all reviewers comments. They are all addressed satisfactory in my opinion. I am looking forward seeing this article in print.

**Have all data underlying the figures and results presented in the manuscript been provided?**

Reviewer #2: Yes

Reviewer #3: Yes

PLOS authors have the option to publish the peer review history of their article (what does this mean?). If published, this will include your full peer review and any attached files.

Reviewer #2: No

Reviewer #3: **Yes: **Benjamin Schwessinger

**Data Deposition**

http://datadryad.org/submit?journalID=pgenetics&manu=PGENETICS-D-20-01371R1

**Press Queries**

---

## [Editor Report · Acceptance letter]

17 Mar 2021

PGENETICS-D-20-01371R1 

Recent loss of the Dim2 DNA methyltransferase decreases mutation rate in repeats and changes evolutionary trajectory in a fungal
pathogen 

Dear Dr Möller, 

We are pleased to inform you that your manuscript entitled "Recent loss of the Dim2 DNA methyltransferase decreases mutation rate in repeats and changes evolutionary trajectory in a fungal
pathogen" has been formally accepted for publication in PLOS Genetics! Your manuscript is now with our production department and you will be notified of the publication date in due course.

With kind regards,

Alice Ellingham

PLOS Genetics

On behalf of:
